# Compressing Context to Enhance Inference Efficiency of Large Language Models

**Yucheng Li[1] , Bo Dong[1] , Frank Guerin[1] , Chenghua Lin[2,3]***

[1] Department of Computer Science, University of Surrey, UK
[2] Department of Computer Science, The University of Manchester, UK
[3] Department of Computer Science, The University of Sheffield, UK
`{yucheng.li, bd00531, f.guerin}@surrey.ac.uk`
`chenghua.lin@manchester.ac.uk`

## Abstract

Large language models (LLMs) achieved remarkable performance across various tasks. However, they face challenges in managing long documents and extended conversations, due to significantly increased computational requirements, both in memory and inference time, and potential context truncation when the input exceeds the LLM's fixed context length. This paper proposes a method called *Selective Context* that enhances the inference efficiency of LLMs by identifying and pruning redundancy in the input context to make the input more compact. We test our approach using common data sources requiring long context processing: arXiv papers, news articles, and long conversations, on tasks of summarisation, question answering, and response generation. Experimental results show that Selective Context significantly reduces memory cost and decreases generation latency while maintaining comparable performance compared to that achieved when full context is used. Specifically, we achieve a 50% reduction in context cost, resulting in a 36% reduction in inference memory usage and a 32% reduction in inference time, while observing only a minor drop of .023 in BERTscore and .038 in faithfulness on four downstream applications, indicating that our method strikes a good balance between efficiency and performance. Code and data are available at `https://github.com/liyucheng09/Selective_Context`.

## 1 Introduction

Large language models (LLMs) have demonstrated remarkable power and impressive generalisation abilities across a wide range of natural language processing tasks, as well as real-life applications (Brown et al., 2020; Touvron et al., 2023; Bubeck et al., 2023). However, a major challenge for existing LLMs is processing longer context. Dealing with longer context with LLMs is fundamen-

---

Context: Large Languages Models (LLMs) have shown their ability to perform new tasks, resulting in a line of work that focuses on further scaling these models. These efforts are based on the assumption {*that more parameters will lead to better performance.*}

Query: What's the assumption behind the efforts to further scale LLMs?

LLMs: Further scaling Large Language Models will lead to better performance on a wide range of tasks.

Figure 1: Some context is redundant because LLMs have learned that knowledge. LLMs can generate the correct answer even when these redundancies are deleted.

tal in scenarios such as having long conversations, document summarisation, and question answering given long documents. However, it is very computationally expensive, particularly with Transformer based LLMs, due to the quadratic growth of memory and computation associated with the 2-D attention matrix (Vaswani et al., 2017). This makes LLMs less accessible and sometimes leads to context truncation during inference. Moreover, due to the above limitation, existing LLMs were usually pre-trained with fixed-context windows, which further constrains their capability in processing longer context.

There are active attempts in reducing the computation and memory cost of the Transformer architecture with sparse attention (Child et al., 2019) or local dense attention (Beltagy et al., 2020). There are also efforts to learn soft prompts with further distillation to save context cost during inference (Mu et al., 2023; Chevalier et al., 2023). In contrast to existing approaches that primarily focus on architectures or distillations, we introduce a fresh perspective to tackle the redundancy in the input context itself, thus proposing a complementary, model-agnostic approach that can be potentially combined with other architecture optimisation methods to further enhance inference efficiency.

---

* Corresponding author

The proposed method is motivated by the potential redundancy and repetition in human language, which has two main sources. The first is the inherent redundancy of natural language. For example, in the conversation *"A: Did you get the chance to pick up groceries today?", "B: Yes, I did get the groceries."*, the underlined part can be seen as a common redundancy in communication. Linguistic studies suggest redundancy is ubiquitous in language (Wit and Gillette, 1999). The other type of input redundancy is from the overlap with training material. As the example in Fig. 1 shows, if some parts of input have already been included in the pre-training stage of LLMs, then it is safe to delete them and the model can still generate the correct answer. In summary, redundancy in the input context, while beneficial for human comprehension, can be superfluous for LLMs and might lead to unnecessary computational expense.

In this paper, we propose *Selective Context*, which prunes redundant content in a given input context, thereby reducing the computational cost and making better use of the fixed context length in LLMs. *Selective Context* evaluates informativeness of lexical units (i.e., tokens, phrases, or sentences) with self-information (Shannon, 1948) computed by a base causal language model. By selectively retaining content with higher self-information, our method provides a more compact and efficient context representation for LLMs to process without compromising their performance on various applications.

We evaluate the effectiveness and different settings of *Selective Context* on arXiv papers, BBC News, and real conversation on ShareGPT.com with four NLP tasks: summarisation, question answering, original context reconstruction, and conversation. Experimental results demonstrate that our proposed method can significantly enhance context efficiency of LLMs during inference while maintaining comparable performance compared to that achieved when full context is used.

## 2 Self-Information

Self-information, also known as *surprisal* or *information content*, is a fundamental concept in information theory that quantifies the amount of information conveyed by an event given a distribution (Shannon, 1948). In the context of language modelling, the event can be regarded as one step of generation (i.e., a token) and the distribution

corresponds to its output distribution. So the self-information of a token can be defined as the negative log likelihood:

$$I(x) = -\log_2 P(x_t|x_0, x_1, ..., x_{t-1}) \quad (1)$$

where $I(x)$ represents the self-information of token $x$ and $P(x)$ denotes its output probability.

In information theory, self-information measures the level of surprise or uncertainty associated with an event; rare events convey more information and thus have higher self-information, while common events convey less information and have lower self-information. In the context of language modelling, self-information can be used to assess the informativeness of lexical units, e.g., words, phrases, or sentences. Lexical units with lower self-information are less informative and thus are more likely to be inferred from the context. As a result, we may treat these parts of input as redundant during LLM inference.

In NLP, self-information has been used to measure surprise in creative language artefacts (Bunescu and Uduehi, 2022). In addition, related concepts of self-information such as entropy and perplexity are widely used in language model optimisation and evaluation.

$$H(S) = \frac{1}{N}\Sigma_t I(x_t) \quad (2)$$

$$PP(S) = 2^{H(S)} \quad (3)$$

where the entropy $H(S)$ of the sentence $S = (x_0, ..., x_n)$ is the average self-information of words in the sentence, and perplexity $PP(S)$ of the sentence can be calculated with entropy. The property of self-information that is especially relevant to our method is the additivity.

$$\begin{aligned} I(x_0, x_1) &= -\log_2 P(x_0, x_1) \\ &= -\log_2 P(x_0)P(x_1|x_0) \\ &= -\log_2 P(x_0) - \log_2 P(x_1|x_0) \\ &= I(x_0)I(x_1) \end{aligned} \quad (4)$$

This means we can calculate the self-information of a lexical unit by simply summing the self-information of the tokens in it.

## 3 Method

*Selective Context* optimises the input context by filtering out redundant or non-essential content to reduce computational cost and make better use of the limited context window. In implementation,

| | |
|---|---|
| **Original:** | INTRODUCTION Continual Learning ( ~~CL~~ ) ~~, also known as Lifelong Learning , is~~ a promising learning paradigm to design models ~~that~~ have to ~~learn~~ how ~~to perform multiple tasks~~ across ~~different environments~~ over ~~their lifetime [~~ To uniform the language and enhance ~~the readability of the paper we~~ adopt the unique term continual learning ~~( CL ). ].~~ Ideal CL models in ~~the real world~~ should ~~be~~ deal ~~with~~ domain shifts ~~,~~ researchers ~~have~~ recently started ~~to~~ sample tasks ~~from~~ two different datasets ~~.~~ For instance ~~,~~ proposed to train and evaluate ~~a model~~ on Imagenet first ~~and then~~ challenge ~~its performance on the~~ Places365 ~~dataset .~~ considers more scenarios ~~,~~ starting ~~with~~ Imagenet or Places365 ~~, and then moving on to~~ the VOC/CUB/Scenes datasets ~~.~~ Few works propose more advanced scenarios built ~~on~~ top ~~of~~ more than two datasets ~~.~~ |
| **Filtered:** | INTRODUCTION Continual Learning ( a promising learning paradigm to design models have to how across over To uniform the language and enhance adopt the unique term continual learning Ideal CL models in should deal domain shifts researchers recently started sample tasks two different datasets For instance proposed to train and evaluate on Imagenet first challenge Places365 considers more scenarios starting Imagenet or Places365 the VOC/CUB/Scenes datasets Few works propose more advanced scenarios built top more than two datasets |

Figure 2: A visualisation of selective context. Darker colour indicates larger value of self-information.

we first 1) employ a causal language model such as GPT (Radford et al., 2019; Brown et al., 2020), OPT (Zhang et al., 2022), or LLaMA (Touvron et al., 2023), computing self-information for each token. We then 2) merge tokens, along with their corresponding self-information values, into lexical units, which can be phrases or sentences. This step is optional if tokens are being used as the basic units. Finally, 3) we eliminate content that is deemed least necessary to render the input more compact.

## 3.1 Computing Self-Information

Given a context $C = x_0, x_1, ..., x_n$, where $x_i$ denotes a token, we use a base language model $M$ to compute the self-information for each token $x_t$ as follows:

$$I(x_i) = -\log_2 P(x_i|x_0, x_1, ..., x_{i-1}) \quad (5)$$

## 3.2 Merging into Lexical Units

If the content filtering of selective context is directly performed on the token level, it might lead to very disjoint context. Therefore apart from token level filtering, we also conduct the filtering procedure on phrase and sentence level. We call a basic unit in our filtering a *lexical unit*, which could be a token, a phrase or a sentence in our setting.

To enable selective context to work on phrases and sentences, we merge tokens and their self-information into lexical units. Each lexical unit $u$ consists of multiple tokens $(x_t, ..., x_{t+\alpha})$, and we can calculate its self-information by summing the self-information of its individual tokens according

to the additivity property of self-information:

$$I(u) = \sum_{i=t}^{\alpha} I(x_i) \quad (6)$$

The NLTK sentence tokenizer is employed to obtain sentence level lexical units. And we use spacy[1] to merge tokens into noun phrases. We do not merge verb phrases as it might produce very long phrases.

## 3.3 Selective Retention of Informative Context

With the self-information of each lexical unit computed, we can now evaluate their informativeness. Instead of using a fixed threshold or retaining a fixed number of top $k$ lexical units, we design a percentile-based filtering approach to adaptively select the most informative content.

First, we rank the lexical units based on their self-information values in descending order. Then, we compute the $p$-th percentile of self-information values among all lexical units.

$$I_p = \text{np.percentile}([I(u_0), .., I(u_k)], p) \quad (7)$$

Next, we selectively retain lexical units with self-information values greater than or equal to the $p$-th percentile, constructing a filtered context $C'$:

$$C' = U_i \mid I(U_i) \geq I_p, 1 \leq i \leq n \quad (8)$$

The percentile-based filtering is a more flexible approach to retain the most informative content depending on the distribution of self-information values in the given context. In Figure 2, we present

---
[1] https://spacy.io/api/pipeline-functions#merge_noun_chunks

an example on phrase level where $p$ is set to 50, which means half of phrases are filtered out. In this case, the context after processing by selective context only retains 57.2% of tokens, which saves 42.8% of context length.

## 4 Experiments

The goal of Selective Context is to reduce the redundancy in the input context without compromising the generation quality of LLMs. As a result, we are expecting the answers given both selective context and the original context to be as close as possible. We take the generated answer given full context as the reference answer, and compare to the generated answer given the selective context in our experiments.

### 4.1 Datasets

Selective Context prunes redundancy in the input context to allow very long context processing for LLMs. However, existing benchmarks for LLMs, such as MMLU (Hendrycks et al., 2020) and ARC (Clark et al., 2018), are mostly single round question answering and are thus not suitable to evaluate our proposed method. Therefore, we collect three test sets consisting of long documents and conversations to evaluate Selective Context. Statistics in detail are presented in Table 4.

**BBC News:** A dataset containing news articles collected from the British Broadcasting Corporation (BBC). This dataset covers a wide range of topics, including politics, business, sports, and technology. We use the full content of each news article in our experiments.

**arXiv Articles:** A dataset consisting of latest academic papers, spaning various scientific disciplines, such as computer science, physics, and mathematics. As arXiv articles can be quite long, we only process the first two sections (usually introduction and background) for each paper in our experiments.

**ShareGPT.com:** ShareGPT.com is a platform where ChatGPT users share their surprising and interesting conversation with ChatGPT. This dataset consists of conversations in different languages and in various scenarios (e.g., coding, chitchat, writing assistant, etc.). We use the ShareGPT dataset for the conversation task in our experiments.

The three evaluation datasets were created carefully to avoid *data contamination*. Data samples in the BBC News, arXiv, and ShareGPT.com datasets were all created after March 2023, which is after

the release of all LLMs in our experiments. Considering some of baseline models are continually being updated, we employ the latest version released before 30 March 2023 to make sure models have never seen our test set in their pre-training and fine-tuning stage. In addition, as some of LLMs in our experiments have a `max_length` of 2048 tokens, we do not include articles or conversations exceeding this length.

### 4.2 Models

We test Selective Context on the following models:
**GPT-3.5 and GPT-4:** GPT-3.5 also known as Chat-GPT, which is likely to be further fine-tuned from GPT-3 and InstructGPT. GPT-4 is the latest model from OpenAI, which has demonstrated substantially improved capability on complex reasoning compared to its predecessor. GPT-3.5 and GPT-4 are unfortunately not open-source, we can only access these models via web api[2].
**LLaMA-7B, 13B, 30B:** LLaMA is a family of open-source language models released by Meta, which is reported to outperform GPT-3 with less parameters. The LLaMA family includes models with size ranging from 7B to 65B. To investigate the effect of scaling law to Selective Context, we experiment with LLaMA with 7B, 13B, and 30B parameters.
**Vicuna-7B, 13B:** Vicuna (Chiang et al., 2023) is a family of open-source language models instruct-tuned from LLaMA. According to their technical report, Vicuna models perform quite well on a list of multitasking benchmarks.

### 4.3 Tasks and Metrics

We evaluate Selective Context on four tasks:
**Original Context Reconstruction:** Given a compressed context produced by Selective Context, this task aims to evaluate whether models are able to reconstruct the original context. This task assesses how well the filtered context retains the essential information from the original context. In our experiments, the compressed contexts are used as input, and the original contexts are used as reference answers.
**Summarisation:** Given a context, the task is to generate a summary that captures the main points of the document. This task aims to evaluate whether Selective Context affects the overall understanding of models on the input contexts. In

---

[2]https://platform.openai.com/docs/api-reference

our experiments, the input and output are the compressed context and the summaries generated based on the compressed contexts. Summaries based on the *original (full) contexts* are treated as the reference answers.

**Question Answering (QA):** Given a document and a set of questions, the task is to generate answers based on the information available in the document. This task aims to evaluate models' understanding of a specific query. Here we first generate questions and answers based on the original context, where these answers are treated as reference answers, and then ask LLMs to answer these questions with selective context.

**Conversation:** This task is only for the ShareGPT dataset. Given a conversation history and a user query, the task is to generate a response to the query based on the previous conversation history. This task aims to evaluate selective context's performance on conversation. Specifically, we ask LLMs to answer the users' last query of ShareGPT conversation instances with selective context applied on the previous conversation history.

We employ four metrics to assess the performance of our models on the tasks: BLEU, METEOR, ROUGE, and BERTScore. BLEU (Papineni et al., 2002) calculates n-gram precision, which is the proportion of n-grams in the generated text that are also present in the reference text. METEOR (Banerjee and Lavie, 2005) takes additional features such as synonymy, stemming and word order into consideration, which leads to more comprehensive evaluation. ROUGE (Lin, 2004) focuses on how much of the important information in the reference text is present in the generated summary. BERTScore (Zhang et al., 2019) leverages contextualised embeddings from pre-trained language models like BERT, computing the cosine similarity between the generated text and reference text embeddings to capture semantic similarity more effectively than traditional n-gram-based metrics.

As mentioned before, we use the generated answers given the full contexts as the reference answers. When testing the deterministic decoding strategy (`greedy` decoding), we take one single run on full context as the reference answer. When testing the non-deterministic decoding strategy (`temperature = 0.7`), we run multiple times on full context to obtain multiple reference answers to address the randomness in decoding. The metrics are computed based on the set of refer-

ence answers. In our experiment, we set the number of reference answers to 4.

## 4.4 Experimental Settings

We use the smaller base causal language model for self-information computing in our experiments. For the LLaMA family and vicuna family, we employ LLaMA-7B to compute self-information. For the OpenAI family, we use a smaller GPT-3 variant `curie` for self-information computing, which is available on OpenAI web API. In self-information computing, we do not process the entire context at once. This is due to our observation on the tendency of LLMs to give later lexical units lower self-information. Instead, we compute self-information sentence by sentence in our experiments.

In our experiments, we compare the two different dimensions that are adjustable in Selective Context.

**Compression Ratios:** We experiment with different content reduction ratios in Selective Context: 0.2, 0.35, 0.5, 0.65, and 0.8. These ratios determine the proportion of content to be filtered out, allowing us to study the trade-off between efficiency and performance as the amount of retained information varies.

**Lexical Units:** Lexical units are the basic element of content reduction in Selective Context. It can be sentence, phrases, or tokens. As mentioned in §3.2, we remove the redundancy in input context by a specific lexical unit level.

## 5 Results

Except for §5.5, all results of selective context presented are at the phrase level (the optimal).

## 5.1 Overview

In Table 1, we first compare the performance of *Selective Context* against the *Original Context* to see how well Selective Context preserves useful information when reducing context cost. The metrics are averaged across all models mentioned in §4.2. The performance drop is shown in parentheses.

As demonstrated in the table, using Selective Context only leads to a marginal drop when the reduction ratio is set to 0.2 or 0.35, despite it significantly reducing the context cost. The BLEU score drops by only 0.05 when 20% of the content is reduced. And the number is even smaller when it comes to ROUGE-1, where the drop is just 0.03. This indicate a high level of consistency be-

| Method | Ratio | BLEU | METEOR | ROUGE | | | BERTScore | | |
|---|---|---|---|---|---|---|---|---|---|
| | | | | Rouge-1 | Rouge-2 | Rouge-L | Precision | Recall | F1 |
| Original | - | .347 | .496 | .571 | .383 | .471 | .910 | .909 | .909 |
| Selective Context | 0.2 | .295 (.05) | .460 (.04) | .540 (**.03**) | .346 (.04) | .438 (.03) | .905 (.005) | .900 (.009) | .902 (**.007**) |
| | 0.35 | .243 (.10) | .421 (.08) | .504 (**.07**) | .294 (.09) | .396 (.07) | .900 (.010) | .894 (.015) | .897 (**.013**) |
| | 0.5 | .179 (.17) | .362 (.13) | .449 (.12) | .237 (.15) | .344 (.13) | .893 (.018) | .882 (.027) | .887 (.023) |
| | 0.65 | .127 (.22) | .299 (.20) | .391 (.18) | .178 (.21) | .287 (.18) | .885 (.025) | .870 (.039) | .877 (.032) |
| | 0.8 | .070 (.28) | .224 (.27) | .311 (.26) | .122 (.26) | .225 (.25) | .874 (.036) | .852 (.057) | .863 (.047) |

Table 1: Comparing Selective Context to the Original Context with `temperature` set to 0.7.

| Method | Ratio | BLEU | METEOR | ROUGE | | | BERTScore | | |
|---|---|---|---|---|---|---|---|---|---|
| | | | | Rouge-1 | Rouge-2 | Rouge-L | Precision | Recall | F1 |
| Random deletion | 0.20 | 0.437 | 0.578 | 0.666 | 0.503 | 0.566 | 0.892 | 0.909 | 0.899 |
| | 0.35 | 0.360 | 0.514 | 0.629 | 0.423 | 0.502 | 0.879 | 0.895 | 0.886 |
| | 0.50 | 0.283 | 0.443 | 0.576 | 0.346 | 0.432 | 0.867 | 0.881 | 0.873 |
| | 0.65 | 0.210 | 0.378 | 0.522 | 0.279 | 0.371 | 0.855 | 0.868 | 0.860 |
| | 0.80 | 0.156 | 0.314 | 0.450 | 0.219 | 0.310 | 0.843 | 0.853 | 0.847 |
| Selective Context | 0.20 | 0.527 | 0.643 | **0.714** | 0.585 | 0.631 | 0.930 | 0.932 | **0.931** |
| | 0.35 | 0.446 | 0.588 | **0.679** | 0.508 | 0.570 | 0.915 | 0.916 | **0.915** |
| | 0.50 | 0.350 | 0.528 | **0.642** | 0.425 | 0.501 | 0.900 | 0.902 | **0.900** |
| | 0.65 | 0.244 | 0.418 | 0.557 | 0.315 | 0.404 | 0.886 | 0.877 | 0.881 |
| | 0.80 | 0.160 | 0.328 | 0.464 | 0.223 | 0.319 | 0.875 | 0.858 | 0.866 |

Table 2: Comparing Selective Context to the random deletion baseline when using `greedy` decoding.

tween answers given selective contexts and original contexts when the reduction ratio is 0.2. Selective Context also yields impressive results when 35% of the content is reduced, with BERT scores around 0.9 and ROUGE-1 scores over 0.5. The drops become noticeable as the reduction ratio rises to 0.5, where the average BLEU score drops 0.17 and the average ROUGE-1 drops 0.12. A reduction ratio of 0.65 and 0.8 tends to be less valuable, as shown by the 0.18 drop on ROUGE-1 and 0.32 drop on BERTScore-F1.

We then compare *Selective Context* against the *Random* compression baseline as shown in Table 2. We observe that using Selective Context allows LLMs to generate very similar answers to the reference answers (answers given full context) although we significantly reduce the context cost. Selective Context maintains BERTScore-F1 above 0.9 when the compression ratio is 0.5 or lower, which shows a high similarity with the reference answers. ROUGE demonstrates the same trend: ROUGE-1 continues to be above 0.64 and ROUGE-L keeps above 0.5 when the ratio is under 0.5. We also notice that Selective Context is significantly more effective than the random baseline: Selective Context with compression ratio of 0.5 shows a better overlapping with the reference answer than Ran-

| Ratio | #Sorry | Answer len. | Unfaithfulness |
|---|---|---|---|
| Full | 0 | 160.3 | - |
| 0.2 | 0 | 156.5 | .027 |
| 0.35 | 6 | 136.0 | .050 |
| 0.5 | 4 | 140.2 | .038 |
| 0.65 | 19 | 131.2 | .051 |
| 0.8 | 27 | 103.7 | .086 |

Table 3: Faithfulness test on `gpt-3.5-turbo` using selective context.

dom baseline with only 20% content compression.

## 5.2 Faithfulness

To evaluate to what extent selective context affects the faithfulness of the LLMs generated content, we perform manual tests on our question answering results based on the idea of Wang et al. (2020). We evaluate 1000 question-answer pairs (200 for each ratio) with the following procedure: 1) We first extract OpenIE tuples from the answers of selective context, and then 2) manually evaluate whether each tuple is entailed by the reference answer. If the model's answer is "Sorry, I don't know", we treat it as "Sorry" cases and do not consider it as unfaithfulness.

As shown in the Table 3, we find that `gpt-3.5` tends to generate shorter answers or refuses to an-

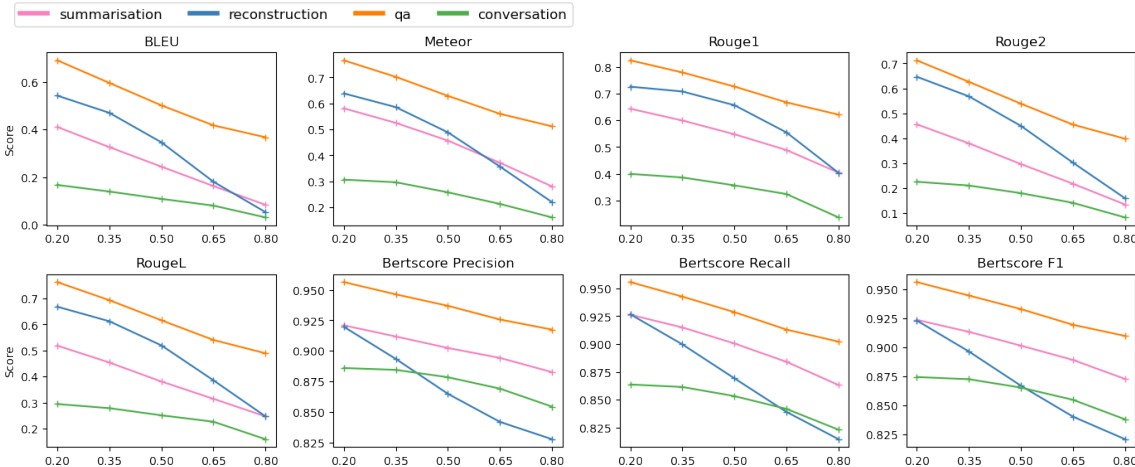

Figure 3: Performance of selective context on different tasks. `x-axix` represents compression ratios (same below).

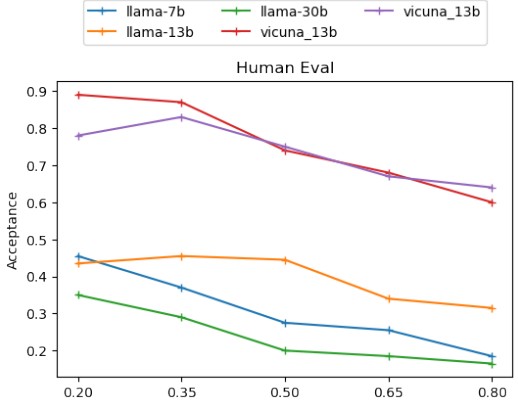

Figure 4: Acceptance rate of generated summaries.

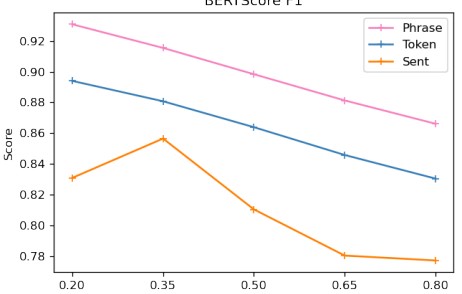

Figure 5: Effects of lexical units.

swer the questions if it fails to identify necessary evidence in the given selective context. With a compression ratio of 0.65, `gpt-3.5` refuses to answer 19 questions (9% of 200), and the answers are 35% shorter than the reference answer (131 tokens in average). However, selective context doesn't significantly affect the faithfulness across all compression ratios. About 3.8% of all tuples are not entailed by the reference answer when the compression ratio is 0.5, and this number rises slightly to 5.1% as the compression ratio increases to 0.65.

## 5.3 Tasks

In this part, we break down and analyse the performances of Selective Context in the four different NLP tasks: summarisation, question answering, original context reconstruction, and conversation. The results are as shown in Fig. 3. First, the results on the Original Context Reconstruction task (RC) show the steepest drop with increasing compression ratio, however, Selective Context allows LLMs to preserve most of the key points in the

original context when the reduction ratio is lower than 0.5, as demonstrated by a rather high ROUGE score. Second, we notice that the curves of question answering and summarisation decrease gradually and are continually higher than those of the other two tasks evaluated by BERTScore. We could say Selective Context is especially suitable for tasks of summarisation and answer generation.

## 5.4 Scaling and Instruct-Tuning

We perform human evaluation to explore the effect of model scales and supervised instruct-tuning on Selective Context. We asked three college students to evaluate 1150 generated summaries from `llama` and `vicuna` (about 55 per model and ratio) and record whether they accept the generation as a reasonable summary. As shown in Figure 4, we find no specific trends between the scales and generation quality given Selective Context. The `vicuna` family demonstrates similar summarisation capability with `7b` and `13b` parameters. And so does the `llama` family, larger models do not show stronger robustness towards Selective Context. But instruct-tuned model `vicuna` demonstrates significantly superior performance

> **Original Context**, `CUDA Memory = 77,695 MB; Time = 110.8 ms/token`
>
> Please see the original document and summary given full context in Appendix B.

> **Selective Context**, Ratio: 0.5, `CUDA Memory = 61,885 MB, Time = 76.3 ms/token, Time to construct selective context = 46.1 ms`
>
> [1]The above paragraph discusses the use of probabilistic methods, safety distance-based control methods, and trajectory prediction methods for assisting vehicles in avoiding collisions. [1]It mentions that CNN has made outstanding contributions in vision tasks, particularly in road condition inspection, due to its excellent regional feature extraction capabilities. However, the performance of CNN-based models for vision tasks is inferior to EfficientNets RepVGG, due to the huge differences between sequential tasks in NLP and image tasks in CV, the difficulty in keeping the original information of inputs after RNN layers, and the computational and memory requirements of switching layers. The paragraph introduces a new network structure called Sequential Spatial Network (SSN) blocks, which overcomes the limitations of traditional CNN-based models. [2]The SSN block consists of convolutional layers, Upsampling Convolutional Decreasing (UCD) blocks, and Reinforcement Region Unit and Fast MultiHead Self-Attention (FMHSA) to enhance local information and improve normalization capability. The paragraph concludes by stating that the SSN network outperforms existing methods on the Lykit dataset and can be easily transferred for image classification tasks.

Figure 6: Comparing the summary generated by `vicuna_13b` given original context and selective context.

than `llama` models given selective context indicating instruct-tuning might help the model to be more robust against context compression. Given selective context, `llama` models often fail to follow instructions and go wild very easily.

### 5.5 Lexical Units

We test the effect of Selective Context based on different lexical units: tokens, phrases, and sentences via BERTScore-F1. As shown in Table 5, employing phrase as the basic lexical units in Selective Context is the optimal approach, consistently outperforming the other two variants, followed by token-level Selective Context. Removing redundancy at sentence-level is a rather unstable implementation compared to the token and phrase-level. This experiment indicates that a reasonable granularity can be crucial in Selective Context.

### 5.6 Case Study

To have a straightforward impression on how well LLMs generate with selective context, we present two summaries given the full and selective context respectively in Figure 6. The original document and processing to obtain selective context are presented in Appendix B.

We first found that preparing selective context is extremely efficient. It takes a one-time cost of 46.1 ms to build selective context for the example paragraph, which includes computing self-information

and performing lexical unit tokenisation. This ensures that the initial stage of establishing a selective context incurs very little overhead. Secondly, it shows selective context significantly reduces the memory usage of the GPU and accelerates the generation process. With a compression ratio of 0.5, selective context reduces about 36% of the memory cost in inference and makes generation 1.32 times faster (per token). By comparing the content of the two summaries, we see that the summary given selective context missed relevant information about the research background (as denoted by the [1] marker), such as the use of machine learning in autonomous driving technology and instead starts with the different methods directly. This is due to the background parts not being selected and removed as redundancy before feeding to `vicuna`. We tried to ask `vicuna`

*"what is the background of this study?"*,
given the selective context, and obtained a decent answer:

*"the research background of this paper is likely to be situated in the domain of autonomous driving technology and the application of artificial intelligence (AI) for improving vehicle safety and decision-making capabilities."*.

This demonstrates that LLMs are likely to be able to infer the deleted parts of background information in the selective context. Selective context also affects `vicuna`'s decision on what informa-

tion should be included in the summary as the second summary includes details about, for example, FMHSA and UCD block (as denoted by the [2] marker) which are not covered in the summary generated with the full context. We find no factual errors in the summary given selective context.

## 6 Conclusion

We introduced Selective Context to improve the context efficiency of LLMs in inference by deleting redundant content measured by self-information. Our extensive experiments on arXiv papers, BBC news articles, and conversation transcripts showed that our proposed method can significantly reduce GPU memory cost, accelerate generation with minor performance decrease, and potentially enable LLMs to handle long documents and extended conversations without the risk of context truncation.

## 7 Limitations

*Selective Context* demonstrates promising results, but it is still necessary to note a couple of potential limitations. Firstly, our approach is somewhat influenced by the phrase boundary detection procedure. We employ the noun phrase tokenisation algorithm provided by `spacy` in our experiments. However, we do not consider verb phrases as there is no mature solution for verb phrase tokenisation. We speculate that we can achieve better compression performance with dependency tree-based filtering procedure which might lead to better boundary identification of lexical units. Secondly, in the experiment section, we use percentile to control the pruning process. However, the optimal compression percentile varies based on specific tasks and context. Developing a tool to find the optimal threshold can further enhance the effectiveness of selective context.

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

## A   Dataset statistics

| Dataset | #Doc | #Sent | #Phrase | #Token |
|---------|------|-------|---------|--------|
| Arxiv | 408 | 28.20 | 514.55 | 864.85 |
| ShareGPT | 470 | 27.35 | 389.42 | 689.32 |
| BBC | 294 | 25.63 | 523.96 | 732.54 |

Table 4: Statistics of the three datasets. #Sent, #Phrase, #Token are averaged per document.

## B   Example of selective context on long context

Here we present an example of selective context on a rather long context. The original paragraphs is from https://arxiv.org/abs/2303.07352. The original paragraphs is shown in Fig. 7. The resulting context is shown in Fig. 8. The reference summary is given in Fig. 9.

## C   The Previous Version of Selective Context

If you're looking for the previous of the paper, please check (Li, 2023).

*INTRODUCTION* In ~~the past many years~~, researchers ~~have~~ ~~focused~~ ~~on~~ how ~~to~~ turn vehicles ~~from~~ assisted driving ~~to~~ more intelligent autonomous driving. Due ~~to~~ the iteration ~~of~~ intelligent hardware ~~and~~ the improvement ~~of~~ chip computing power, ~~a large amount of data~~ collected ~~by sensors can be~~ quickly converted ~~and~~ fed ~~into~~ models ~~to make decisions~~. In the driving process, the safety factor ~~is~~ ~~the first consideration for~~ users ~~and~~ researchers. Therefore, how AV should avoid collisions has ~~become~~ ~~a top priority~~. Concepts such ~~as~~ probabilistic methods ( eg . : ~~Markov chains~~ ~~and~~ ~~Monte Carlo~~ ), safety distance-based control methods , ~~and~~ trajectory prediction methods ~~have~~ ~~been~~ designed ~~in recent years to~~ cope ~~with~~ complex traffic conditions. In terms ~~of~~ vision, CNN ~~has~~ made outstanding contributions and ~~has been~~ applied ~~to a large number of~~ road condition inspection tasks due ~~to~~ its excellent regional feature extraction capabilities. The local feature information obtained ~~by~~ CNN will ~~be used for~~ obstacle detection. Secondly, because the motion trajectory ~~is~~ planned for AV , the relationship ~~between~~ each local feature ~~of the image~~ obtained ~~by~~ CNN needs ~~to be~~ established. Some strategies ~~are~~ based ~~on~~ CNN plus RNN so ~~that they can~~ deal ~~with~~ sequential graphs as input , eg . : STDN . Although the above strategies ~~have~~ performed ~~well in a large number of~~ vision tasks , their performances ~~are still far~~ inferior ~~to similar-sized convolutional neural networks counterparts~~ , ~~such as~~ EfficientNets ~~and~~ RepVGG . We believe this ~~is~~ due ~~to~~ the following aspects . First , the huge differences ~~between~~ the sequential tasks ~~of~~ NLP ~~and~~ the image tasks ~~of~~ CV ~~are~~ ignored . For example , when the local feature information acquired ~~in a two-dimensional image~~ ~~is~~ compressed ~~into one-dimensional time series information~~ , how ~~to~~ achieve accurate mapping becomes ~~a difficult problem~~ . Second , ~~it is~~ difficult ~~to~~ keep the original information of inputs since after RNN layers , ~~we need to~~ recover the dimension from one ~~to~~ three . Besides , due ~~to~~ the several transformations between different dimensions , that process becomes ~~even harder~~ , especially ~~since~~ our input size ~~is~~ 224×224×5 . Third , the computational and memory requirement ~~of~~ switching ~~between~~ layers are extremely heavy tasks , ~~which~~ also becomes a tricky point ~~for the algorithm to run~~ . Higher hardware requirements as ~~well as more~~ running ~~time~~ arise ~~when running~~ the attention part . In ~~this paper~~ , ~~we propose a new network structure based on~~ CNN ~~and~~ attention to vision tasks ~~in autonomous driving~~. The new network structure overcomes ~~these problems by using~~ Sequential Spatial Network (SSN) blocks . As shown ~~in~~ Fig . , input images first go ~~through~~ the convolution stem for ~~fine-grained feature extraction~~ , ~~and are then fed into~~ a stack ~~of~~ SSN blocks ~~for further processing~~ . The Upsampling Convolutional Decreasing (UCD) blocks ~~are~~ introduced for ~~the purpose of~~ local information enhancement ~~by~~ deep convolution , ~~and~~ in SSN block of features generated in ~~the first stage~~ can ~~be~~ less loss of image resolution , ~~which is~~ crucial ~~for~~ the subsequent trajectory adjustment task . In addition , we adopt a staged architecture design using five convolutional layers ~~with~~ ~~different kernel sizes~~ ~~and~~ steps gradually ~~decreasing the resolution~~ ( sequence length ~~)~~ ~~and~~ flexibly ~~increasing the dimensionality~~ . Such a design helps ~~to~~ extract local features ~~of~~ different scales ~~and~~ , since the first stage retains high resolution , our design ~~can~~ effectively reduce ~~the resolution of~~ the output information ~~in~~ ~~the first layer~~ at ~~each convolutional layer~~ , ~~thus reducing the computational effort of subsequent layers~~ . The Reinforcement Region Unit ~~( RRU )~~ and the Fast MultiHead Self-Attention (FMHSA ~~)~~ in the SSN block can help obtain ~~global and local structural information~~ within the intermediate features and improve the normalization capability ~~of the network~~ . Finally , average pooling ~~is used to obtain~~ better trajectory tuning . Extensive experiments ~~on~~ the lykit dataset demonstrate ~~the superiority of~~ our SSN network ~~in terms of accuracy~~. In addition ~~to~~ image classification , SSN block can ~~be~~ easily transferred ~~to other vision tasks~~ and serve ~~as~~ a versatile backbone.

Figure 7: Selective context on the introduction of `https://arxiv.org/abs/2303.07352`

In researchers how turn vehicles assisted driving more intelligent autonomous driving Due the iteration intelligent hardware the improvement chip computing power collected quickly converted fed models In the driving process the safety factor users researchers Therefore how AV should avoid collisions has Concepts such probabilistic methods ( eg . : ) , safety distance-based control methods , trajectory prediction methods designed cope complex traffic conditions In terms vision CNN made outstanding contributions and applied road condition inspection tasks due its excellent regional feature extraction capabilities The local feature information obtained CNN will obstacle detection Secondly because the motion trajectory planned for AV the relationship each local feature obtained CNN needs established Some strategies based CNN plus RNN so deal sequential graphs as input eg STDN . Although the above strategies performed vision tasks their performances inferior EfficientNets RepVGG . We believe this due the following aspects First the huge differences the sequential tasks NLP the image tasks CV ignored For example when the local feature information acquired compressed how achieve accurate mapping becomes Second difficult keep the original information of inputs since after RNN layers recover the dimension from one three Besides due the several transformations between different dimensions that process becomes especially our input size 224×224×5 Third , the computational and memory requirement switching layers are extremely heavy tasks also becomes a tricky point Higher hardware requirements as running arise the attention part In CNN attention to vision tasks The new network structure overcomes Sequential Spatial Network (SSN) blocks As shown Fig , input images first go the convolution stem for a stack SSN blocks The Upsampling Convolutional Decreasing (UCD) blocks introduced for local information enhancement deep convolution , in SSN block of features generated in can less loss of image resolution crucial the subsequent trajectory adjustment task In addition we adopt a staged architecture design using five convolutional layers steps gradually ( sequence length flexibly Such a design helps extract local features different scales , since the first stage retains high resolution our design effectively reduce the output information at The Reinforcement Region Unit and the Fast MultiHead Self-Attention (FMHSA in the SSN block can help obtain within the intermediate features and improve the normalization capability Finally average pooling better trajectory tuning Extensive experiments the lykit dataset demonstrate our SSN network In addition image classification SSN block can easily transferred and serve a versatile backbone .

Figure 8: The resulting context

**Given Original Context**, CUDA Memory = 77,695 MB; Time = 110.8 ms/token

The paragraph discusses the development of autonomous driving technology and the use of artificial intelligence and machine learning to make decisions in the driving process. The safety of the vehicle is a top priority and methods such as probabilistic methods, safety distance-based control methods, and trajectory prediction methods have been developed to cope with complex traffic conditions. The use of convolutional neural networks (CNN) has been successful in image tasks such as obstacle detection, but the performance of these networks is still inferior to similar-sized convolutional neural networks counterparts. This is due to the huge differences between the sequential tasks of NLP and the image tasks of CV, the difficulty of keeping the original information of inputs, and the heavy computational and memory requirement of switching between layers. The authors propose a new network structure based on CNN and attention to vision tasks in autonomous driving, using Sequential Spatial Network (SSN) blocks. The network structure overcomes these problems by using deep convolution and attention mechanisms to extract local and global features, and reducing the computational effort of subsequent layers. The proposed network structure is evaluated on the lykit dataset and shows superior performance in terms of accuracy. B.

Figure 9: The reference summary generated by vicuna-13b given the full context.