# OpenReview forum: "Compressing Context to Enhance Inference Efficiency of Large Language Models"
_EMNLP/2023/Conference — EMNLP 2023 Main_

### Official Review · Reviewer_6QAt · 2023-08-01

**Typos Grammar Style And Presentation Improvements:** Figure 4 mentions vicuna_13b twice. I…
**Soundness:** 4

**Excitement:**

3: Ambivalent: It has merits (e.g., it reports state-of-the-art results, the idea is nice), but there are key weaknesses (e.g., it describes incremental work), and it can significantly benefit from another round of revision. However, I won't object to accepting it if my co-reviewers champion it.

**Paper Topic And Main Contributions:**

This paper introduces an information-theoretic way to compress the input contexts to LLMs. Specifically, it uses the notion of self-information to elide parts of the input that have low information content with respect to the underlying distribution of a language model - a process the authors call "Selective Context". The authors show a number of different experimental results demonstrating the effectiveness of the approach, and argue that Selective Context leads to significant reductions in memory usage and inference time with respect to full inputs, while observing some drops in performance on several downstream applications.

**Questions For The Authors:**

As I understand it, currently the choice of lexical units means that Selective Context eliminates either words, phrases OR sentences. Have the authors considered trying to perform eliminations across lexical units?

There is an implicit assumption that the weaker model used to compute self-attention has the same "knowledge" as the more powerful model used to do inference. This is of course not necessarily true. Have the authors considered using the same model for both Selective Context and inference to establish an upper bound of sorts on performance?

**Reasons To Accept:**

The paper tackles a real and important problem with LLMs today. While context sizes continue to grow, there are considerable computational savings that can be achieved by more efficient representation of the input context. The authors have also proposed an interesting and novel solution that has information-theoretic foundations to the problem context compression. Finally, the paper is generally well written and easy to follow.

**Reasons To Reject:**

My concerns with the paper are mainly related to the experimental section, and it isn't clear to me that the results support the claims that the authors are making.

Firstly, the headline figure in the abstract states a 32% reduction in inference time. However, it is unclear whether this factors in the cost of actually computing the self-information for every token in the input and subsequently compressing it. Given that this computation is also done by an LLM (albeit a less powerful one), this seems like a non-negligeable factor to report.

Further, the paper does not compare Selective Context against a baseline approach that simply uses the weaker model to generate a summary (with a prompt that states the compression ratio, for example). As a result, it is unclear whether this complex process of computing self-information for every token yields any meaningful gains over a naive way of compressing the input context.

Beyond this, there are some unusual choices in the presentation of the experimental results that don't make much sense to me. For example, Figure 4 only presents open-source LLMs (why?); conversely Table 3 only shows results on gpt-3.5turbo; gpt-4 is introduced but is never spoken about again. In similar vein, Tables 1 & 2 show averaged scores instead of breaking up scores by model. I understand the overall message here is to show that the authors' approach is model agnostic. However, for the sake of deeper insight about how different models respond to having input elided, a clearer and more principled approach to showing results would have been appreciated.

**Reproducibility:**

4: Could mostly reproduce the results, but there may be some variation because of sample variance or minor variations in their interpretation of the protocol or method.

**Reviewer Confidence:**

4: Quite sure. I tried to check the important points carefully. It's unlikely, though conceivable, that I missed something that should affect my ratings.

---

> ### Author Rebuttal · Authors · 2023-08-28
>
> Response 3:
> Thanks for your thoughtful reading and constructive comments for our paper!
> To respond your questions:
>
> **“It’s unclear whether the time cost of the context compression is included”**, many thanks for raising this question. I should have made that more clear in the paper. Time used to compress context is already included in the paper. Actually, computing self-information is highly efficient. It only requires one step to compute self-information and it is very easy to compute it in parallel. According to section 5.6, it only takes 46 ms to finish context compression. Then it leads to a 35 ms reduction for every token the LLM generates. For the example in Figure 6, where the answer has about 400 tokens, we actually save 14000 ms in total. In summary, the 36% reduction did factor the cost of computing self-information. We will emphasise this in our final version.
>
> **“Why not directly ask a model to summarise the input?”**, this is absolutely a very interesting idea, and we believe this is an interesting baseline to be compared against. However, the limitation of this idea is obvious: the generation of a summary can be very slow (hundreds times slower than self-information computing). As mentioned above, self-information is a super-efficient approach that only requires one-step to calculate. In addition, we have very little control over the generation of summaries. For example, we cannot ask the model to generate a summary that is exactly 20% shorter than the input. On the contrary, our method enables a fine-grained adjustment based on compression ratio. In summary, using a model to summarise the input might be a great idea for text compression, but it would not help accelerating the inference of LLMs.
>
> **“Why are GPT models missed in figure 4?”**. Figure 4 illustrates how model scale and instruct-tuning affects the overall performance. However, as GPT models are closed models, we have little knowledge on their scales and whether or how they were instructed-tuned. Therefore, we cannot add GPT models in Figure 4.
>
> **“Why is only gpt-3.5-turbo analysed in Table 3”**. We chose to conduct a manual evaluation on gpt-3.5-turbo specifically because it is the only large language model for which previous studies have analysed faithfulness. And we use their approach (B Li et al., 2023, H Zhang et al., 2023) in our experiments. This is why our analysis focuses solely on gpt-3.5-turbo. We didn’t perform human evaluation on all baseline models as the annotation procedure can be quite costly, and we believe the results of gpt-3.5-turbo are representative.
>
> **“Tables 1 & 2 should present separate scores for different models”**, first many thanks for your clear understanding of  our intention of using average scores to show our approach is model agnostic. We didn’t present separate scores for all baseline models to avoid the table taking up too much space and ensure the table is readable. However, we do agree that it is very valuable to provide more detailed results, and we will add a new table presenting full breakdown results in the Appendix in the final version.
>
> At last, we would like to share an interesting new experiment of our method conducted on the recently released LongEval that assesses model performance on long context processing. The results suggest our method is robust while dealing with extremely long context.
>
> |Method|Accuracy|GPU memory usage|Inference time per step|
> |---|---|---|---|
> |our method (50% reduction) + gpt-3.5-8k| .990 | unknown | unknown |
> |gpt-3.5-16k | .992 | unknown | unknown |
>
> |Method|Accuracy|GPU memory usage|Inference time per step|
> |---|---|---|---|
> |our method (50% reduction) + vicuna-4k-v1.5-7b| .968 | 68,433 MB |  81.8 ms/token |
> |vicuna-16k-v1.5-7b | .938 | 123,874MB |  130.8 ms/token |
>
> Thank you again for your comments.

---

### Official Review · Reviewer_eKVf · 2023-08-03

**Soundness:** 4

**Excitement:**

3: Ambivalent: It has merits (e.g., it reports state-of-the-art results, the idea is nice), but there are key weaknesses (e.g., it describes incremental work), and it can significantly benefit from another round of revision. However, I won't object to accepting it if my co-reviewers champion it.

**Paper Topic And Main Contributions:**

The paper proposes a methodology to quantify importance of spans of tokens in a context so that we can remove the least important tokens from context and still maintain reasonable performance while gaining speed up from the reduced length of the context.

The proposed method uses the generation likelihood of tokens, which translates to the self information of the tokens to quantify the important of the tokens.

The paper proposes using percentiles (instead of thresholds) to select the top set of tokens to be kept in context.

With an extensive set of experiments, the authors show that even when dropping significant amount of tokens from context, the model is still able to perform very well on a wide variety of tasks.

Since the method performs best if the input has redundant information, the paper also proposes three datasets with long text and conversations to evaluate performance.

**Questions For The Authors:**

Please look at "Reasons to Reject" section. Would love to see your opinion on each of the points mentioned there.

**Reasons To Accept:**

1. Making efficient inference with long text in context is a very important area of research that can bring down compute used for these tasks significantly.
2. The method is simple, and works with prominent LLMs, and is very well verified by extensive set of experiments.

**Reasons To Reject:**

1. The method may not generalize very well on tasks that specifically focus on text being dropped.
2. Even though the method reduces the latency of the LLM, there is significant overhead in calculating the tokens that should be dropped.
3. If we combine the latency from finding the tokens to be dropped, and the latency of the LLM, the overall latency gain isn't very significant while the quality loss might be larger.
4. Is it possible to quantify the quality loss better on some non-generative task like NLI, so that we have a better sense of how much the quality loss is with metrics like F1.
5. There are other methods that are more sophisticated and offer better guarantees. For examples see [1] and [2]:
  - [1] Dynamic Context Pruning for Efficient and Interpretable Autoregressive Transformers. (https://arxiv.org/pdf/2305.15805)
  - [2] PromptCast: A New Prompt-based Learning Paradigm for Time Series Forecasting (https://arxiv.org/abs/2210.08964)

**Reproducibility:**

5: Could easily reproduce the results.

**Reviewer Confidence:**

4: Quite sure. I tried to check the important points carefully. It's unlikely, though conceivable, that I missed something that should affect my ratings.

---

> ### Author Rebuttal · Authors · 2023-08-28
>
> Response 2:
> Thanks for your thoughtful reading and constructive comments for our paper!
> Really appreciate your approval of our method and experiments.
> We respond to your main questions below:
>
> **“Method may not work when tasks are focusing on text being dropped”**. This is a great question. We have actually addressed this question in the Case Study as well as in Figure 1. These results suggest that the context compression mainly targets the redundancy of input, where the dropped text is either 1) learned by the model in pretraining; or 2) has been mentioned before in the input. In Figure 1, even though the query is focusing on the dropped text, LLMs are still able to generate the correct answer thanks to their inner-knowledge.
>
> **“Computing self-information will lead to significant overhead”**, We should have made ourselves clearer in the paper. Actually, computing self-information is highly efficient. It only requires one step to compute self-information and it is very easy to compute it in parallel. As shown in section 5.6, it only takes 46 ms to finish context compression. Then it leads to a 35 ms reduction for every token the LLM will generate. For the example in Figure 6, where the answer has about 400 tokens, we actually save 14000 ms in total. In summary, self-information computing is highly effective compared to the answer generation.
>
> **“Using NLI benchmarks can allow us to have a better sense on quality loss”**, This is a very interesting idea. But since NLI questions are usually very short (which means there’s less space for further compression) and their answers are usually one single label, the efficiency of generation is not considered as a problem. But your suggestion reminds us that maybe we could try a SQuAD fashion test that uses exact match as the criterion, then we can check the accuracy that better quantifies quality change.
>
> **“Other sophisticated methods”**, many thanks for kindly providing valuable references. I have gone through the two papers. Both of them seem to accelerate LLMs inference via involving extra trainable parameters. We believe our approach complements the studies you mentioned, as it is effective and can be integrated with other methods without conflict.
>
> In addition, we would like to share an interesting new experiment of our method conducted on the recently released LongEval that assesses model performance on long context processing. The results suggest our method is robust while dealing with extremely long context.
>
> |Method|Accuracy|GPU memory usage|Inference time per step|
> |---|---|---|---|
> |our method (50% reduction) + gpt-3.5-8k| .990 | unknown | unknown |
> |gpt-3.5-16k | .992 | unknown | unknown |
>
> |Method|Accuracy|GPU memory usage|Inference time per step|
> |---|---|---|---|
> |our method (50% reduction) + vicuna-4k-v1.5-7b| .968 | 68,433 MB |  81.8 ms/token |
> |vicuna-16k-v1.5-7b | .938 | 123,874MB |  130.8 ms/token |
>
> Thank you again for your comments.

---

### Official Review · Reviewer_PgsA · 2023-08-04

**Soundness:** 2

**Excitement:**

2: Mediocre: This paper makes marginal contributions (vs non-contemporaneous work), so I would rather not see it in the conference.

**Paper Topic And Main Contributions:**

This paper proposes a method of identifying and pruning redundancy in the input context to enhance the inference efficiency and reduce memory cost considering the limited context window of LLM. Evaluations on common long texts dataset and multi-tasks show that it can maintain comparable performance while achieving memory and cost reduction to a great extent.

**Questions For The Authors:**

See more in Reasons To Reject sections.


**Reasons To Accept:**

* The topic of the paper is crucially important for researchers in LM's and AI/AGI with limited context window and large computation cost with Transformer model.
* This work helps to solve the intrinsic problem (redundancy and repetition) in human language in LLM generation.
* Compared with existing solutions focusing on architectures or distillations, it proposes a light filter on input without changing or finetuning the model.

**Reasons To Reject:**

1. Missing references and discussions about related works. There is already some work but the current paper includes very few discussions about existing similar efforts.
2. The phrase and sentence level filtering is mostly related to the boundary detection and how tokens are merged.  Inappropriate combinations will lead to the misunderstanding of filtered texts and bad performance in downstream tasks. Methods here are not delicately designed. Besides, no further discussion on the impact when using various methods.
3. Dataset used in the experiments are too short for long context processing. Please refer to the well acknowledged datasets that are recently proposed for long context evaluations.

**Reproducibility:**

4: Could mostly reproduce the results, but there may be some variation because of sample variance or minor variations in their interpretation of the protocol or method.

**Reviewer Confidence:**

4: Quite sure. I tried to check the important points carefully. It's unlikely, though conceivable, that I missed something that should affect my ratings.

---

> ### Author Rebuttal · Authors · 2023-08-28
>
> Response 1:
> Thanks for your thoughtful reading and constructive comments for our paper!
> To respond your questions:
>
> **“Missing references and discussions”**, we have included popular attempts to accelerate LLMs inferences in the paper such as sparse attention (Child et al., 2019), and soft prompt (Mu et al., 2023; Chevalier et al., 2023). But we do notice other relevant efforts like rectified position embedding (https://github.com/bojone/rerope) and dynamic attention masking (Anagnostidis et al. 2023). We will definitely include more related works in the final version to fully address the reviewer's concern given the extra page. We would also appreciate if the reviewer can share with us what references we might have missed, as indicated in his/her comment.
>
>
> **“the boundary detection is not delicately designed”**, we adopt spaCy, a NLP toolkit, to decide the boundary of phrases in the paper, which is a standard approach. We appreciate the suggestions regarding developing more delicate boundary detection and testing various methods. However,  the primary focus of this work is to demonstrate the potential of context pruning, rather than focus on boundary detection. In addition, we found spaCy was sufficient and did a good job in our experiments. Despite the basic boundary detection, our approach still demonstrated strong performance on accelerating LLMs inference and maintaining good generation quality.
>
> **“Dataset is too short for long context processing”**, that is a great suggestion. We have noticed LongEval recently released by lmsys.org that aims to evaluate long context processing. So we test our method on the line retrieval task with two LLMs. Note that we use a slightly different way to compute self-information due to the long context. As shown in the following table, the results suggest our method is robust while dealing with extremely long context.
>
> |Method|Accuracy|GPU memory usage|Inference time per step|
> |---|---|---|---|
> |our method (50% reduction) + gpt-3.5-8k| .990 | unknown | unknown |
> |gpt-3.5-16k | .992 | unknown | unknown |
>
> |Method|Accuracy|GPU memory usage|Inference time per step|
> |---|---|---|---|
> |our method (50% reduction) + vicuna-4k-v1.5-7b| .968 | 68,433 MB |  81.8 ms/token |
> |vicuna-16k-v1.5-7b | .938 | 123,874MB |  130.8 ms/token |
>
> We will add the LongEval experiments as a complementary part in the final version.

---

### Meta-Review · Area_Chair_7WjS · 2023-09-08

**Recommendation:** 4

**Metareview:**

This paper presents an efficient inference method that ranks the spans of tokens in the context, and removes the least important ones. The reviewers appreciate the importance of the problem, the simplicity, novelty and theoretical foundation of the solution, and the writing quality. Nonetheless, there were some concerns about missing references and comparisons, and missing ablations, though many of them did not include a specific missing comparison, so they can be discounted. There were also practical gains given the method overhead. The rebuttal seems to have addressed most of the concerns raised.

---

### Decision · Program_Chairs · 2023-10-07

**Decision:**

Accept-Main

**Comment:**

This paper presents an efficient inference method that ranks the spans of tokens in the context, and removes the least important ones. The reviewers appreciate the importance of the problem, the simplicity, novelty and theoretical foundation of the solution, and the writing quality. Nonetheless, there were some concerns about missing references and comparisons, and missing ablations, though many of them did not include a specific missing comparison, so they can be discounted. There were also practical gains given the method overhead. The rebuttal seems to have addressed most of the concerns raised.